# Canine Cytokines Profile in an Endemic Region of *L. infantum*: Related Factors

**DOI:** 10.3390/vetsci9060305

**Published:** 2022-06-20

**Authors:** Pablo Jesús Marín-García, Lola Llobat

**Affiliations:** Department of Animal Production and Health, Veterinary Public Health and Food Science and Technology (PASAPTA), Facultad de Veterinaria, Universidad Cardenal Herrera-CEU, CEU Universities, 46115 Valencia, Spain; pablo.maringarcia@uchceu.es

**Keywords:** *Leishmania infantum*, infection, cytokines, purebred, crossbred, endemic region

## Abstract

Canine leishmaniosis is caused by infection with parasite *Leishmania infantum*, which are transmitted by sandflies *Phlebotomus.* Canine leishmaniosis is an endemic disease in the Mediterranean region. The immune response could vary between hosts and determines the severity of the disease and clinical features. The aim of this study was to analyze the serum levels of cytokines TNF-α, IFN-γ, IL-2, IL-6, and IL-8, which are related to the activation of Th1 or Th2 immune responses in dogs living in the *L. infantum* endemic region. Moreover, we intend to relate and correlate these levels with different factors, such as sex, age, diet, lifestyle, and breed. Epidemiological data and serum were recovered for seventy-eight dogs, and serum levels of cytokines described previously were analyzed by using the ELISA method. The results showed differences in serum levels of IFN-γ, IL-2, and IL-8 between breeds. The lifestyle also affected serum levels of IL-2. The main conclusion of this study is that Ibizan hounds and crossbred dogs have a serological profile of cytokines that seems to indicate certain protections against infection by *L. infantum* compared to boxer and purebred breeds.

## 1. Introduction

Leishmaniosis is a zoonotic disease caused by infection with the obligate intracellular protozoan parasite *Leishmania* spp. that are transmitted by the phlebotomine sandflies from the Psychodidae family, and the genus *Phlebotomus* is its main vector in the Mediterranean region [1,2,3]. This parasitosis could be presented in mucocutaneous or visceral form, and the latter is the most pathogenic and is caused by the species *Leishmania infantum*, which is predominant in the Mediterranean area [4,5,6]. Although this parasite can infect different types of mammals, including wild animals and cats [7,8], the domestic dog (*Canis lupus familiaris*) is considered the principal reservoir of leishmaniosis due to *L. infantum* in the Mediterranean basin, Middle East, and South America [5,9]. In Spain, the high prevalence of seroprevalence of *L. infantum* infection in dogs has been demonstrated in the Mediterranean zone, mainly the Balearic Islands, and Valencian Region [10]. However, not all infected dogs develop the disease [3,11,12].

The immune response produced after infection by *L. Infantum* could vary between hosts and determines the severity of the disease [13,14,15]. Once the parasite has been inoculated by the sandfly’s bite, the host macrophages present the *L. infantum* antigen to undifferentiated Th0 lymphocyte [4,16]. *L. infantum* infection actives adaptative immune responses, with a balance between Th1 and Th2 response. The activation of Th1 lymphocytes triggers a cellular response with the activation of macrophages, the synthesis of free oxygen radicals, and the destruction of the parasite, and this response is related with controlling infections [17,18]. The cytokines involved in this pathway are interleukin 2 (IL-2), tumor necrosis factor alpha (TNF-α), and interferon gamma (IFN-γ) [4,15,16]. This response makes it possible to control the infection, and it is known as a protective response, since the host does not show clinical signs of the disease [19]. In fact, elevated serum levels of IFN-γ are related with low parasite counts and few or no clinical signs compared to sick dogs [20], and these levels increase with anti-*Leishmania* treatment together with clinical improvement and low blood parasitemia [21,22]. Infected dogs exhibit increases in serum levels of IFN-γ, IL-6, and IL-18 and decreases in TNF-α, IL-2, and IL-8 compared to non-infected dogs [23].

Different factors could affect the immune response of dogs. Several studies indicate that nutritional supplements could modify Th1 inflammatory responses [24,25]. In humans, it has been demonstrated that aging affects mainly adaptative immune response, changing cytokines levels and inflammatory responses [26,27,28]. Although studies in this regard in dogs are not as abundant, some studies indicate that aging dogs experience changes like human in inflammation and immune function [29,30]. However, one of the most relevant factors that seems to affect the immune response in *Leishmania* infection is the canine breed. In fact, the prevalence of infection differs between canine breeds in endemic regions of *L. infantum* [31], which could indicate immunological protection of some canine breeds compared to others against infection by the parasite and, mainly, the development of leishmaniosis. Several studies show that the activation of Th1 immune response depends on the canine breed such that some breeds present different serum levels of cytokines [22,32,33]. In fact, different studies suggest a certain resistance and/or susceptibility to clinical diseases in dogs related to breed. In this sense, our research group recently published a study where a clear difference was observed in the prevalence of leishmaniosis according to the canine breed, presenting a higher prevalence in breeds such as boxer or Doberman (evolutionarily related) in which the prevalence of clinical signs reaches up to around 50% [31]. This difference in the prevalence of leishmaniosis could be indicating an underlying cause of genetic origin that could be related to different immune responses, or with the activation of different immunity pathways. Solano-Gallego et al. (2000) demonstrated that Ibizan hound dogs have a higher cellular immune response than other breeds and suggested that the Ibizan hound is a special population regarding immune responses against *Leishmania* infection [34]. Sanchez-Robert et al. (2008) conducted a study with nineteen canine breeds, including Ibizan hounds and boxer breeds, where they analyzed the presence of polymorphisms in the Slc11a gene, which is linked to susceptibility toward canine visceral leishmaniosis [35].

The aim of this study is to relate the serum levels of different cytokines in dogs living in endemic *L. infantum* regions with different factors that could affect these levels, including canine breeds.

## 2. Materials and Methods

### 2.1. Ethics Approval

The experiments involving animals were conducted according to the guidelines of the Declaration of Helsinki and approved by the Animal Experimentation Ethics Committee of the Universidad Cardenal Herrera CEU, with code 2020/VSC/PEA/0216.

### 2.2. Animals and Data Collection

Seventy-eight dogs living in the Valencia Community region (East of Spain, Mediterranean region) were used for the study, and the serum samples and epidemiological data were recovered from October 2021 to May 2022. For all animals, the following epidemiological data were recovered: sex (two categories: male or female); age (four categories: puppies—less than one year old; young—between one and five years old; adult—between five and ten years old; elder—more than ten years old); breed (four categories: purebred, Ibizan hound, boxer, or crossbred); with official vaccination in order or not; with or without leishmaniosis vaccination; with or without the use of external deworming and type; lived with other animals or not; type of feeding; indoor or outdoor animals; with *L. infantum* infection detected and, if yes, symptomatic or asymptomatic animals.

### 2.3. Samples Collection and Cytokines Levels

Ten milliliters of blood samples were collected from cephalic vein with Vacutainer tubes without anticoagulants to recovery serum samples. Serum was obtained and preserved at −20 °C before analysis. Serological testing for *L. infantum* detected specific antibodies using the indirect immunofluorescent antibody test (IFAT), which was conducted by an external laboratory, and IFAT for anti-*Leishmania*-specific immunoglobulin G (IgG) antibodies (MegaFLUO LEISH, Megacor Diagnostik GmbH, Hörbranz, Austria). Dog serum was considered seropositive with IFAT titer ≥ 1:80, following the manufacturer’s instructions [36,37].

To recover serum, all samples recovered without anticoagulants were maintained at room temperature and were centrifugated to obtain serum aliquots. Serum levels of TNF-α, IFN-γ, IL-2, IL-6, and IL-8 were measured by a commercial kit of indirect ELISA assay (Canine TNF-α ELISA kit, Canine IFN-γ, Canine IL-2 ELISA kit, Canine IL-6 ELISA kit, and Canine IL-8 ELISA kit; Invitrogen; and Canine IL-18 ELISA kit, MyBiosource, respectively) following the manufacturer recommendations. Briefly, a 50–100 µL of serum was added to the microplate wells. Then, a biotinylated detection antibody specific for different cytokine and Avidin-Horseradish peroxidase (HRP) conjugates were added successively to each microplate well and incubated. Free components were washed away. The substrate solution was added to each well. The enzyme-substrate reaction was determined by using optical density (OD), and it was measured spectrophotometrically at a wavelength of 450 nm. The serum concentration of different cytokines was calculated by comparing the OD of the samples to the standard curve.

### 2.4. Statistical Analysis

Epidemiological data were recovered, and serum levels of cytokines were analyzed using the general linear model procedure (PROC GLM) of the statistical package SAS (North Carolina State University, USA), after normality and homoscedasticity were tested by Shapiro–Wilks and Levene tests, respectively. The model was carried out with sex, age, type of feed, vaccinated against *Leishmania* or not, lived with other animals or not, and breed as fixed effects. Pearson’s correlations between cytokine levels were carried out. The statistical significance was set at *p*-value < 0.05.

## 3. Results

The epidemiological data recovered are showed in Table 1. Briefly, all animals were outdoor, and most of the animals were owner dogs (91.03%): 51.28% of animals used were males, and 10.26% were puppies (less than one year old), whereas 33.33% were elder (more than ten years old). Most dogs lived with other animals (88.46%), and these animals included other dogs and some tested dogs lived in the same household and were fed commercial diets (88.46%). A percentage of 91.03% with respect to dogs used external deworming, and most dogs have the correct official vaccination, whereas only 10.26% were vaccinated against *Leishmania* (Table 1).

Only 14.10% of dogs were positive for *L. infantum* infection by ELISA antibody detection, and 81.82% had at least one clinical sign. All animals with clinical signs presented parasitic burden above 1:100. The clinical signs found, and their prevalence are shown in Table 2.

Table 3 shows the data on the main cytokines controlled in this work. By using a very diverse population (Table 2), the data set used in this study showed a wide range. No expression of TNF-α was found.

Serum levels of cytokines have only been related to breed and living or not with other animals. Animals that do not live with other animals presented higher serum levels of IL-2 than animals that live with other animals (+51%; *p* = 0.0073) (Figure 1).

Significant differences were also observed according to boxer breed, Ibizan hound, and other purebreds, and crossbreds presented statistical differences in IFN-γ, IL-2, and IL-8 serum levels (Figure 2).

None of the Ibizan hounds were infected. The serum levels of IFN-γ were similar in boxer, elder animals, and infected animals, and they were lower than Ibizan hounds (*p* < 0.0001). No differences between infected animals and other groups were found in the serum levels of analyzed cytokines.

## 4. Discussion

This study analyzed the canine serum levels of cytokines TFN-α, IFN-γ, IL-2, IL-6, and IL-8 in endemic *L. infantum* regions with different factors such as age, sex, diet, lifestyle, and breed in seventy-eight dogs.

Out of the animals examined, 14.10% were positive to *L. infantum* infection, in accordance with data observed in other reported studies on Mediterranean areas, where prevalence was estimated between 5 and 57.1% [38]. Previous studies in indoor animals estimated this prevalence in 4.74% in other Mediterranean areas [31], and these higher levels were found probably because the animals used in this study were all outdoor. Of the positive animals, 81.82% presented values above 1:100 of parasitic burden and had at least one clinical feature. Other studies in experimental infected dogs indicated a clinical state in 66% [23], whereas a recent meta-analysis found that only 43% of all animals presented cutaneous and clinical changes [39]. Differences between the prevalence could be due to the parasitic burden quantification technique. Although the Office International des Epizooties recommend the detection for serological methods and the use of parasitological or molecular methods for confirmation in the case of lower titers [24,40], several studies indicate that molecular methods are more sensitive and, concretely, real-time PCR for diagnosis of leishmaniasis is the method chosen in humans (see review [41]). In fact, several studies realized in Brazil and Venezuela recommend the confirmation of leishmaniosis in dogs using molecular methods [42,43]. Related to serological methods, ELISA seems to present higher sensitivity and specificity than IFAT [44]. However, IFAT continues to be the method of choice in many studies. Quantitative serological methods such as ELISA or IFAT determine antibody levels and can confirm the infection of compatible clinical signs, but low antibody levels require confirmation by molecular techniques, according to the recommendation of LeishVet guidelines [12]. However, the most common clinical practice is the detection only using serological methods, mainly the IFAT method.

In agreement with our results, studies realized in dogs naturally infected in Brazil showed that around 90% of dogs infected presented clinical signs [45]. The most common clinical features were lymphadenomegaly (50%) and conjunctivitis (30%), and they are firstly related to high parasitic burden. These results agree with results obtained previously in other studies in Spain [22,46] and with results in experimental infected dogs, where the authors found 50% of lymphadenomegaly and 25% of conjunctivitis [23]. In natural infected animals, the results are similar, with 55.9% of lymphadenopathy found by Freitas et al. (2012) in Brazil. In agree with our results, these authors found hyperglobulinemia. However, they indicated a reduction in red blood cells and hematocrit, unlike the results in this study where the most common biochemical alterations were proteinuria and hyperglobulinemia (30%) [45].

Related to serum cytokines levels, TNF-α is not expressed in the analyzed animals. On the other hand, different authors showed high levels of TFN-α and its expression in skin, lymph node cells, and spleen and liver extracts in animals with visceral leishmaniosis, and these were related to parasitic burden [47,48,49,50]; only Do Nascimento et al. (2013) related the serum levels of TNF-α with parasitic burden in the spleen, indicating that the levels of TNF-α increases with spleen parasitism [51], whereas De lima et al. (2007) did not observe statistical differences in the serum levels of TNF-α in infected dogs with or without visceral symptoms [52]. In our study, the number of parasites in the spleen has not been analyzed; therefore, it is possible that the parasite burden in our animals was so low that serum TNF-α levels were undetectable by the ELISA method. Therefore, the quantification of parasitic burden in spleen and detection of TNF-α by other methods that are more sensitive than PCR would be interesting.

No differences were found in serum cytokines levels related to sex, age, and diet. In humans, several studies were carried out to analyze the sex differences of immune responses. The results were that male monocytes produce higher levels of IL-12, IL-1, and TNF-α and lower levels of IL-2 to lymphocytes than females, which suggests a relative suppression of the cellular immune response of the specific immune system in men [53]. However, according to our results, no differences were found in serum levels of IFN-γ, IL-2, IL-6, and IL-8 that were related to sex in dogs [54,55]. In humans, some studies indicated that the elevated age results in decreased activities in the immune system [28] and in the response to vaccination [56], which is also shown in dogs [57]. However, human studies have shown that this difference is due to the age-associated accumulation of adipose tissue, which is the cause of differences in inflammatory cytokine levels [58,59]; thus, leptin induces the production of TNF-α, IL-6, and IL-12 in obese human and mouse [60,61]. Given that none of our animals were obese or overweight, it is to be expected that there would be no differences in the levels of cytokines analyzed, even though they were of different ages. Although some studies indicate the relationship between immune response and diet [56], only two studies linked certain nutritional supplements with a different immune response in dogs [25,62]. Concretely, Cortese et al. (2015) showed that nutraceuticals supplementation diet could regulate the immune response to *L. infantum* infection in dogs [25].

Serum levels of IL-2 were higher in dogs living with other animals than those not living with other animals. IL-2 has a key role in the regulation of immune response and is essential for the maturation of Treg [63,64], and it could be a regulator for Th1/Th2 balance [65]. In fact, this cytokine regulates T cells, and it is fundamental to starting immune responses [66]. In serum dogs immunized with *Lutzomyia longipalpis* salivary protein LJM17, high levels of IL-2 have been found in dogs vaccinated against *Leishmania* [67], and IL-2 is the most abundant in the serum of cured and asymptomatic dogs than uninfected dogs [68]. The serum levels of IL-2 found in animals that live with other animals could indicate an exposure of these animals to the parasite or even a very early infection, which is still undetectable by IFAT.

The high levels of IL-2 have also been found in Ibizan hound dogs and low levels in Boxer breeds, whereas the groups of purebred and crossbred dogs presented intermediate values. Abbehusen et al. (2018) show that high levels of IL-2 and IL-6 are present in immunized dogs [67]. However, no differences in serum levels of IL-6 were found between breed groups. Similar results were observed by Pinelli et al. (1994) in a study examining infected and non-infected dogs [19], whereas other authors indicate high levels of IL-6 in infected animals, suggesting a mixture of inflammatory and anti-inflammatory responses [23]. The levels of IL-6 have been related to disease progression in visceral leishmaniasis [69,70]. Therefore, no differences in the serum levels of IL-6 found in animals studied could be related to the low number of animals with clinical signs included in our study and with low levels of disease progression observed. Similarly to IL-2, serum levels of IFN-γ were higher in Ibizan hound dogs than boxer and purebred, whereas the crossbred presented intermediate levels. Some studies have demonstrated that high IFN-γ could act as a protective mechanism against infections [32], which could indicate more protection against *L. infantum* infection in Ibizan hounds and crossbred than boxer and purebred. Recently, it has been demonstrated in *L. infantum*-infected dogs that purebred dogs present more frequent clinical features than infected crossbred dogs [71]. On the contrary, IL-8 was lower in Ibizan hounds than boxer and purebred dogs, presenting crossbreed intermediate serum levels. The levels of IL-8 have been found to be lower in uninfected than natural and experimentally infected dogs of *L. infantum* [23], and neutrophils from healthy dogs produced higher levels of IL-8 than neutrophils from *Leishmania*-infected dogs [72].

These results could indicate an immunological protection mainly in the Ibizan hound, and to a lesser extent in crossbred dogs. Our results show that different immune responses occur depending on the dog’s breed, mainly in Ibizan hound and boxer, according to several studies [31,34,35].

## 5. Conclusions

The prevalence of *L. infantum* infection was 14.10%. Of the infected animals, 81.82% presented values above 1:100 of parasitic burden and at least one clinical feature. The most common clinical signs were lymphadenomegaly (50%) and conjunctivitis (30%), whereas the most common biochemical alterations were proteinuria and hyperglobulinemia (30%). The serum levels of cytokines were not affected by the sex, age, and diet of the dog. However, animals that lived with other animals presented higher serum levels of IL-2 than animals that have not lived with other animals. The most relevant factor that affected serum cytokine levels was canine breed. The Ibizan hound showed higher levels of IL-2 and IFN-γ and lower levels of IL-8 than boxer and other purebred animals, suggesting differentiated immune responses in these two canine breeds. Crossbred presented intermediate values of IFN-γ, IL-2, and IL-8. These results could indicate a different immune response depending on canine breed. Cytokine mRNA expression studies in whole blood samples and genetic studies from different canine breeds could be interesting for verifying whether serum cytokine values correspond to differences in gene expression. More studies including other cytokines and other canine breeds are necessary to elucidate the immune response pathway in different genetic animals.

## Figures and Tables

**Figure 1 vetsci-09-00305-f001:**
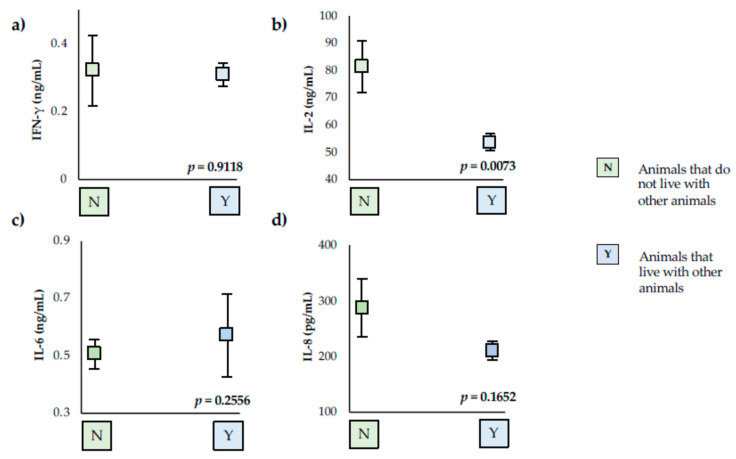
Serum levels of the cytokines studies in animals that do not live with other animals (N) and that live with other animals (Y). (**a**) Interferon gamma (IFN-γ), (**b**) Interleukin 2 (IL-2), (**c**) Interleukin 6 (IL-6), and (**d**) Interleukin 8 (IL-8). Squares represent mean values for each group analyzed, and vertical lines represent standard deviation. The values for IL-8 are pg/mL, and for IFN-γ, IL-2 and IL-6, they are ng/mL. Each cytokine value possessing different letters differs significantly (*p* < 0.05).

**Figure 2 vetsci-09-00305-f002:**
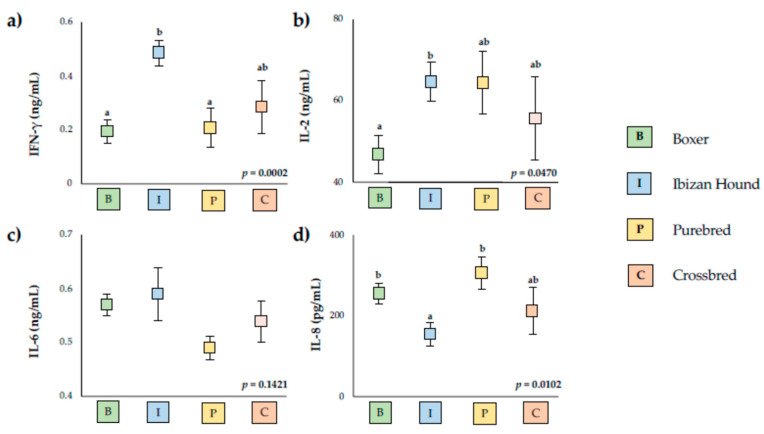
Serum levels of the cytokines studies in boxer (B) and Ibizan hound (I) and another purebred (p) and crossbred (C). (**a**) Interferon gamma (IFN-γ), (**b**) Interleukin 2 (IL-2), (**c**) Interleukin 6 (IL-6), and (**d**) Interleukin 8 (IL-8). Squares represent mean values for each group analyzed, and vertical lines represent standard deviation. The values for IL-8 are pg/mL, and for IFN-γ, IL-2, and IL-6, they are ng/mL. Each cytokine value possessing different letters differs significantly (*p* < 0.05).

**Table 1 vetsci-09-00305-t001:** Epidemiological data recovered of the animals studied.

Variable	Categories	No. of Dogs (%)
Dog population	Households	71 (91.03)
	Animal shelters	7 (8.91)
Gender	Male	40 (51.28)
	Female	38 (48.72)
Age	Puppy (<1 year)	8 (10.26)
	Young (1 to 5 years)	21 (26.92)
	Adult (5 to 10 years)	23 (29.49)
	Elder (>10 years)	26 (33.33)
Breed	Ibizan hound	28 (35.90)
	boxer	31 (39.74)
	Purebred (no Ibizan hound and boxer)	28 (35.90)
	Crossbred	7 (8.97)
Diet	Commercial	69 (88.46)
	Home prepared/raw food consumption	9 (11.54)
Lived with other animals	Yes	69 (88.46)
	No	9 (11.54)
Obligatory vaccination	Yes	69 (88.46)
	No	9 (11.54)
Anti-Leishmania vaccination	Yes	8 (10.26)
	No	70 (89.74)
External deworming	Yes	71 (91.03)
	No	7 (8.91)
Overall		78 (100.00)

**Table 2 vetsci-09-00305-t002:** Clinical features and anti-*Leishmania* antibodies titer observed in positive animals and its prevalence.

Clinical Feature	No. of Dogs (%)	Anti-Leishmania Antibodies Titer
Lymphadenomegaly	5 (50)	1:640 to 1:1280
Weight loss	2 (20)	1:640
Lethargy	1 (10)	1:1280
Pale mucous membranes	2 (20)	1:640
Splenomegaly	1 (10)	1:100
Exfoliative dermatitis	2 (20)	1:640
Onychogryphosis	1 (10)	1:100
Blepharitis	1 (10)	1:640
Conjunctivitis	3 (30)	1:200
Thrombocytopenic anemia	1 (10)	1:100
Leukocytosis/leukopenia	1 (10)	1:200
Thrombocytopenia	2 (20)	1:200
Hyperglobulinemia	3 (30)	1:200 to 1:1280
Hypoalbuminemia	2 (20)	1:200
Reduced ALB/GLOB ratio	2 (20)	1:200
Proteinuria	3 (30)	1:200
Renal azotemia	1 (10)	1:200
Hepatic transaminase elevation	2 (20)	1:640
Overall	10 (100)	1:100 to 1:1280

**Table 3 vetsci-09-00305-t003:** Ranges of different cytokines analyzed. The table shows the number of data analyzed (*n*), range, mean ± standard deviation (SD), and coefficient of variation (CV).

Cytokine ^1^	*n*	Range ^2^	Mean ± SD ^2^	CV (%)
IFN-γ	72	0.1–1.29	0.31 ± 0.03	88
IL-2	75	10.37–148.00	56.71 ± 3.01	46
IL-6	70	0.57–0.01	0.37 ± 0.95	20
IL-8	77	0–624.00	223.00 ± 17.15	67

^1^ IFN-γ: interferon gamma; IL: interleukin. ^2^ The values for IL-8 are expressed pg/mL, and for IFN-γ, IL-2, and IL-6, they are expressed ng/mL.

## Data Availability

Not applicable.

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
