# Peer review of "Canine Cytokines Profile in an Endemic Region of L. infantum: Related Factors"

_vetsci, 2022, doi:10.3390/vetsci9060305_

Round 1

Reviewer 1 Report

Canine cytokines profile in an endemic region of L. infantum: 2 related factors , is an interesting article. The authors

showed differences in serum levels of IFN-ϒ, IL-2, and IL-8, between breeds, levels that can be influenced by the dogs lifestyle.

  1. The introduction is not sufficient. Although authors introduced with backgrounds of the parasite Leishmania infantum, and the detection techniques. There are no clear scientific questions raised by the authors based on this background.

  1. The authors did not discuss the limitations of the methods

  1. Reformulate paragraph 136-138

  2. Regards

Author Response

We welcome the reviewer's suggestions and comments. We are sure that he has improved the quality of the manuscript remarkably. We answer the questions raised one by one in the attached document.

Reviewer 2 Report

In the presented manuscript, Marin-Garcia and Llobat analyzed the levels of different cytokines in the sera of dogs that live in the Leishmania infantum endemic region. It was found that the levels of IL-2 and IFN-gamma depended on the canine bred, and were enhanced in Ibizan hound and crossbred dogs compared to boxer and purebred dogs. The authors suggested that the  increased levesl of IL-2 IFN-gamma may contribute to the protection against L. infantum. I have the following comments:

Lines 115-116 – it would be better to clarify in this sentence that some tested dogs lived in the same household.  

How many dogs of each breed (pure-bred, Ibizan hound, boxer and crossbred) have been tested ?

Table 3 – Could you please explain somewhere in the text why the number (n) of analyzed data is lower than 78?

Have the levels of interleukins in the sera of vaccinated and nonvaccinated dogs been compared ?

Lines 250-262 – Please consider whether this information can be presented in the introduction section?

Author Response

(The authors gave the same response as above.)

Reviewer 3 Report

The manuscript entitled " Canine cytokines profile in an endemic region of L. infantum: related factors” analyzed the serum levels of cytokines TNF-α, IFN-ϒ, IL-2, IL-6, and IL-8, related to activation Th1 or Th2 immune response in dogs lived in L. infantum endemic region.

The study is interesting but not very innovative.  I recommend making some changes before considering publication. Diagnosis should not be based only on serological tests. it is necessary to assess parasite load in all dogs under study by molecular investigations.

Also:

introduction -add bibliography on the endemic role of the region being studied.

-line 35. write "l. infantum" in italics.

-line 121. what method was used to detect anti-Leishmania antibodies? clarify this point.

Finally, the investigation Cytokine mRNA expression in whole blood samples from dogs is necessary to support the conclusions obtained.

Author Response

(The authors gave the same response as above.)

Round 2

Reviewer 3 Report

Accept in present form